# Resilience, Stress, and Burnout Syndrome According to Study Hours in Spanish Public Education School Teacher Applicants: An Explanatory Model as a Function of Weekly Physical Activity Practice Time

**DOI:** 10.3390/bs12090329

**Published:** 2022-09-11

**Authors:** Eduardo Melguizo-Ibáñez, Gabriel González-Valero, José Luis Ubago-Jiménez, Pilar Puertas-Molero

**Affiliations:** Department of Didactics of Musical, Artistic and Corporal Expression, University of Granada, 18071 Granada, Spain

**Keywords:** burnout syndrome, stress, resilience, physical activity, elementary education

## Abstract

The selection process for the teaching profession in public elementary education is difficult, which can lead to the appearance of disruptive states in applicants. For this reason, the present study aimed to establish the relationship between study hours and the levels of stress, burnout, and resilience in applicants to the Spanish public teaching profession. Accordingly, this objective was achieved by (a) developing an explanatory model of study hours according to levels of stress, burnout, and resilience, and (b) contrasting this model through a multigroup analysis according to whether students performed more than 3 h of physical activity per week. A descriptive, comparative, cross-sectional study was carried out on a sample of 4117 applicants (31.03 ± 6.800), using an ad hoc socio-demographic questionnaire, the Perceived Stress Scale, the Maslach Burnout Inventory, and the Connor–Davidson Resilience Scale for data collection. The results revealed that participants who practiced more than 3 h of physical activity per week showed lower levels of stress and burnout syndrome, manifesting higher levels of resilience. Furthermore, better associations between resilience and the other constructs were also observed for people who practiced more than 3 h of physical activity per week. In conclusion, the practice of physical activity can help to decrease stress and develop key elements for the selective exam of the Spanish public teaching corps.

## 1. Introduction

In Spain, the established process for becoming a teacher in the state’s public teaching corps consists of passing a rigorous and demanding competitive examination [1]. So-called “*opositores*” are the applicants for the various posts in the Spanish state’s public teaching corps, who can obtain their place after sitting the necessary exams. The objective of the competitive examination phase is to demonstrate specific knowledge of the teaching specialty being applied to [1], consisting of two tests: (1) a practical test to determine whether the candidates or opponents possess the scientific ability and mastery of the technical skills corresponding to the specialty for which they are applying, together with the written development of a topic chosen at random by the selection board; (2) a defense of the teaching program referring to the curriculum of the area within the specialty being applied to, together with the preparation and presentation of a teaching unit by the applicant based on the official syllabus [1].

Obtaining a place in the public teaching profession manifests high levels of uncertainty and fear in the applicants, due to the stress generated by the entrance exam [2]. The onset of stress occurs in three phases [3]. The first of these is the “alarm reaction”, through which the subject is warned to be alert to a certain situation [4], during which the stress levels sometimes do not rise [5]. If this phase is prolonged, the resistance phase is reached, where the person tries coping with the situation, but realizes that there is a limit to their endurance capacity; thus, they may become frustrated [6]. This is followed by the exhaustion phase, which is characterized by fatigue and may also affect the emotional and motivational level [7].

A continuous subjection to stress levels, together with disruptive thoughts, can generate a state known as burnout [8]. This concept is defined as a phenomenon characterized by a loss of motivation, emotional exhaustion, physical fatigue, and low levels of tolerance and commitment to a task [9]. It can also be characterized by emotional exhaustion, low self-fulfillment, and depersonalization [10]. Furthermore, it has been observed that continuous subjection to these levels of burnout and stress can develop negative effects on a mental and physical health, reflecting detrimental symptoms such as a continuous state of tiredness, muscle fatigue, and psychological disorders [11].

Resilience plays a fundamental role in overcoming the above-described disruptive effects; this concept was defined by San Román-Mata et al. [12] as the intrinsic capacity of human beings to overcome stressful and adverse situations in order to achieve their goals. Contextualizing this term to the academic environment, resilience plays a fundamental role in achieving academic goals, with factors such as self-efficacy, planning, stress control, and persistence playing a key role [13,14]. Another key factor that helps to achieve the proposed objectives is the practice of physical activity [12]. This concept can be defined as a bodily movement of any kind produced by muscle contraction that results in a substantial increase in energy expenditure in the individual [15]. Physical activity triggers the release of neurotransmitters such as serotonin and dopamine, which help to improve concentration and to channel disruptive states generated by the academic environment [16]. In this case, it has been shown that the practice of physical activity together with high levels of resilience can help to improve academic performance and, therefore, attitude when facing an assessment test [17].

Within the field of physical sports, research has been carried out covering the proposed study variables, but previous studies have not focused on this study’s particular population. Zurita-Ortega et al. [18] concluded that the practice of physical activity helps to increase the levels of resilience in judokas, helping them to face tasks with greater motivation and to overcome disruptive states more quickly. Within the academic field, González-Valero et al. [19] affirmed that burnout syndrome can be alleviated by the effects of resilience, but an active lifestyle together with emotional intelligence can help to reduce the effects of burnout syndrome and the stress generated in the process of preparing for academic tasks [20].

In light of the above, the following research hypotheses are proposed:

**H1.** 
*Participants who engage in more than 3 h of physical activity per week will show a positive relationship between resilience and study hours.*


**H2.** 
*Participants who engage in more than 3 h of physical activity per week will show a negative relationship between study hours and burnout and stress levels.*


**H3.** 
*Participants who engage in less than 3 h of physical activity per week will show a negative relationship between study hours and resilience.*


**H4.** 
*Participants who engage in less than 3 h of physical activity per week will show a positive relationship between study hours and increased levels of burnout and stress.*


Therefore, the current research is aimed at establishing the relationship between study hours and stress levels, burnout, and resilience in Spanish public teacher candidates. This objective was achieved by (a) developing an explanatory model of study hours, stress levels, burnout, and resilience, and (b) contrasting this model through a multi-group analysis according to whether students performed more than 3 h of physical activity per week.

## 2. Materials and Methods

### 2.1. Design and Participants

A descriptive, nonexperimental (ex post facto), cross-sectional design was used for this study. The sample consisted of a total of 4117 participants (31.03 ± 6.800), of whom 1363 (33.1%) were male and 2754 (66.9%) were female. Likewise, taking into account the sampling error, an error of 5.0% was established for a confidence level of 99% for the size of the study population.

### 2.2. Instruments

**An ad hoc socio-demographic questionnaire** was implemented, aimed at collecting sociodemographic variables such as sex and age of the participants. This instrument was also used to determine whether the participants were physically active (Do you do more than 3 h of physical activity per week?), offering a dichotomous response (yes/no) [21,22,23]. It was also used to collect the number of hours of study per day (How many hours do you study per day?).

**The Perceived Stress Scale** [24] was also implemented, using the Spanish version [25] in this study. This questionnaire is composed of a total of 14 items with a five-point Likert scale response format (0 = never, 1 = almost never, 2 = occasionally, 3 = often, and 4 = very often). In this case, the instrument obtained a high degree of reliability, with a Cronbach’s alpha score of 0.879. The reliability for the entire scale had a score of α = 0.898.

**The Maslach Burnout Inventory** (**MBI**) [26] was also implemented, using the Spanish version [27] in this study. This questionnaire is composed of 22 items, which are subdivided into three scales. The first is related to emotional exhaustion and consists of nine items (1, 2, 3, 6, 8, 13, 14, 16, and 20). The second assesses the degree of depersonalization and consists of five items (5, 10, 11, 15, and 22). The third subscale measures the degree of personal fulfillment through eight items (4, 7, 9, 12, 17, 18, 19, and 21). Regarding the degree of reliability, the first subscale had a score of α = 0.807, the second had a score of α = 0.881, and the third had a score of α = 0.852. The degree of reliability for the whole scale had a score of α = 0.888.

**Lastly, the Connor–Davidson Resilience Scale** (**CD-RISC**) [28] was applied, using the Spanish version [29] in this study. This scale is composed of 25 items, where resilience is assessed from a multidimensional perspective: persistence/tenacity/self-efficacy (items 10, 12, 16, 17, 23, 24, and 25), control under pressure (items 6, 7, 14, 15, 18, 19, and 20), adaptability and support networks (items 1, 2, 4, 5, and 8), control and purpose (items 13, 21, and 22), and spirituality (items 3 and 9). In this case, the first dimension of persistence/tenacity/self-efficacy obtained a score of α = 0.846. The dimension related to control under pressure scored α = 0.723. The dimension related to adaptability and support networks scored α = 0.794. The dimension related to control and purpose showed a high degree of reliability with α = 0.908, while the dimension related to spirituality scored α = 0.796. The degree of reliability for the whole scale scored α = 0.924.

### 2.3. Procedure

Initially, a systematic review was carried out in order to understand the problems addressed in this research. Once conclusions were drawn from the review, a Google Form was created with the instruments described above, together with the different objectives of this research. Due to the COVID-19 pandemic, the questionnaire was shared via telematic means due to the mobility restrictions imposed. To ensure that the questions were not answered randomly, two questions were duplicated. Accordingly, 43 questionnaires were deleted as they were not correctly filled in. With respect to the ethical principles governing the research, the present study followed the principles established in the Declaration of Helsinki, guaranteeing the anonymity of each participant and treating the data for scientific purposes. This research was approved and supervised by the ethics committee of the University of Granada (1230/CEIH/2020).

### 2.4. Data Analysis

For the statistical analysis of the results, the IBM SPSS Statics 25.0 program (IBM Corp, Armonk, NY, USA) was used. A Student’s *t*-test for independent samples was carried out, determining statistically significant differences by means of the Pearson chi-square test, establishing the significance level at 95%. To calculate statistical power, Cohen’s standardized *d* [30] was used. In this case and depending on the value obtained, the effect was classified as null (≤0.19), small (0.20–0.49), medium (0.50–0.79), and strong (≥0.80). The normality of the sample was determined using the Kolmogorov–Smirnov test, obtaining an adequate degree of normality.

The statistical software IBM SPSS Amos 26.0 program (IBM Corp, Armonk, NY, USA) was used to develop the structural equation models. In this case, the models allowed studying the differences between those who practiced more than 3 h of physical activity per week and those who did not. Each model was composed of a single exogenous variable (RES) and eight observed or endogenous variables (PER, CUP, ADP, CP, SPR, NSH, BURN, and STR). For the latter variables, the causal relationships were examined by focusing on the observed associations between the degree of reliability of the different measurements and the different causal relationships, in such a way as to allow for inclusion of the error. In terms of direction, the unidirectional arrows originate from the different regression weights and symbolize the lines of influence. Significance levels of 0.05 and 0.001 were also established. In this case, the different models focused on the influence of resilience, stress, and burnout syndrome on study hours.

For the evaluation of the model, the criteria established by Bentler [31] and McDonald [32] for the different parameters were followed. These authors indicated that the goodness of fit should be evaluated according to the chi-square test, with nonsignificant *p*-values indicating a good fit of the model. The values of the comparative fit index (CFI), goodness-of-fit index (GFI), and incremental reliability index (IFI) should be higher than 0.900, while the root-mean-square error of approximation (RMSEA) values should be lower than 0.100 to obtain a good fit (Figure 1).

## 3. Results

The comparative analysis of the results revealed that the participants who met the physical sports criteria manifested lower levels of stress (STR) (M = 33.94) than those who did not (M = 37.40). Continuing with the variables that constitute burnout syndrome, higher scores of emotional exhaustion (EE) (M = 38.61) and depersonalization (DP) (M = 16.94) were observed for people who did not practice more than 3 h of physical activity per week; however, those who fulfilled this criterion obtained higher scores of personal fulfillment (PR) (M = 25.20). Continuing with the variables that constitute resilience, subjects who practiced more than 3 h of weekly physical activity showed higher scores in control and purpose (CP) (M = 2.65), control under pressure (CUP) (M = 2. 79), adaptability and support networks (ADP) (M = 2.58), and spirituality (SP) (M = 2.36), whereas participants who did not meet these physical sport criteria showed higher levels of persistence/tenacity/self-efficacy (PER) (M = 2.65) (Table 1).

The model developed and proposed for participants who practice more than 3 h of physical activity per week obtained good values for each of the indices evaluated. The chi-square test obtained a nonsignificant *p*-value (χ^2^ = 1.528; df = 3; pl = 0.007). Despite good results, these data cannot be interpreted in isolation due to the statistical sensitivity of the sample size [33]. The comparative fit index (CFI), normalized fit index (NFI), incremental fit index (IFI), and Tucker–Lewis index (TLI) obtained values of 0.977, 0.933, 0.979, and 0.935, respectively. Furthermore, the root-mean-square error of approximation (RMSEA) showed a value of 0.053.

Table 2 and Figure 2 show the values obtained for participants who practiced more than 3 h of physical activity per week. Regarding resilience (RES), a negative relationship was observed with stress (STR) (*r* = −0.549; *p* ≤ 0.001), burnout syndrome (BURN) (*r* = −0.162; *p* ≤ 0.001), and control under pressure (CUP) (*r* = −0.399; *p* ≤ 0.001). On the contrary, positive relationships of this variable were observed with the number of hours of study (NHS) (*r* = 0.131; *p* ≤ 0.001), persistence/tenacity/self-efficacy (PER) (*r* = 0.860), adaptability and support networks (ADP) (*r* = 0.890; *p* ≤ 0.001), control and purpose (CP) (*r* = 0.827; *p* ≤ 0.001), and spirituality (SP) (*r* = 0.272; *p* ≤ 0.001). Stress (STR) showed a positive relationship with burnout syndrome (*r* = 0.337; *p* ≤ 0.001) and the number of hours of study (*r* = 0.085; *p* ≤ 0.05), while a negative relationship was observed between the number of study hours (NHS) and burnout syndrome (BURN) (*r* = −0.074; *p* ≤ 0.05).

The proposed model for participants who did not practice more than 3 h of physical exercise per week showed a good fit for each of the different indices. The chi-square test obtained a nonsignificant *p*-value (χ^2^ = 1.946; df = 4; pl = 0.000). The comparative fit index (CFI), normalized fit index (NFI), incremental fit index (IFI), and Tucker–Lewis index (TLI) obtained values of 0.915, 0.903, 0.905, and 0.900, respectively. Furthermore, the root-mean-square error of approximation (RMSEA) showed a value of 0.069.

Figure 3 and Table 3 show the results obtained for participants who practiced less than 3 h of physical activity per week. In this case, for resilience (RES), a negative relationship was observed with stress (STR) (*r* = −0.637; *p* ≤ 0.001), burnout syndrome (BURN) (*r* = −0.098; *p* ≤ 0.001), and control under pressure (CUP) (*r* = −0.531; *p* ≤ 0. 001), while a positive relationship was observed with persistence/tenacity/self-efficacy (PER) (*r* = 0.857), adaptability and support networks (ADP) (*r* = 0.880; *p* ≤ 0.001), control and purpose (CP) (*r* = 0.822; *p* ≤ 0.001), spirituality (SP) (*r* = 0.165; *p* ≤ 0.001), and number of study hours (NSH) (*r* = 0.113; *p* ≤ 0.001). For the stress variable (STR), positive relationships were observed with the number of study hours (NHS) (*r* = 0.094; *p* ≤ 0.05) and burnout syndrome (r = 0.489; *p* ≤ 0.001). Lastky, a negative relationship was shown between the number of study hours (NHS) and burnout syndrome (*r* = −0.044).

## 4. Discussion

This discussion aims to provide an explanation for the results obtained. For this reason, the results obtained are compared with those of other research studies.

Comparative analysis revealed that participants who practiced more than 3 h of physical activity per week showed lower levels of stress than those who do not meet this criterion. Given these results, it has been argued that regular sports practice helps to channel disruptive states and emotions that are detrimental to people’s health [34]. Likewise, the type of physical activity also influences the development of stress levels, since, for high-intensity interval training, an increase in anxiety and perceived stress was also observed [35], due to the activation of the sympathetic nervous system together with the secretion of neurotransmitters such as epinephrine and norepinephrine [36]. A sedentary lifestyle also impairs people’s mental self-image, thus leading to higher levels of stress related to body dissatisfaction [37].

Regarding the effect of sports practice on burnout syndrome, it was observed that participants who practiced more than 3 h of physical activity showed lower scores in emotional exhaustion and depersonalization, with this group showing higher scores in personal fulfillment. The results obtained in the present study are supported by numerous studies [38,39], stating that the regular practice of physical exercise brings numerous benefits at a mental level [40,41,42], due to the secretion of neurotransmitters such as serotonin and dopamine [16]. It was also found that lifestyles that are not very active from a physical sports point of view are related to the presence of a negative personal image [43]. In this case, the practice of sport can help to channel the levels of burnout syndrome; however, the motivation toward the practice of physical activity can help to increase or decrease these levels [40]. The study carried out by Sánchez-Alcaraz Martínez and Gómez-Marmol [44] concluded that the high number of hours per week that high-performance athletes train affects the prevalence of burnout syndrome, increasing it when the proposed objectives are not achieved.

Concerning resilience, it was observed that, for all its component variables except for persistence, participants who practiced more than 3 h of physical activity per week showed higher scores. Given these findings, sport can be used as a means of coping with different negative or disruptive states [9]. Likewise, sport practice allows for the development of multiple capacities, such as a positive mentality, optimism, the development of emotional intelligence, the acceptance of adverse situations, and the development of motivational factors for overcoming disadvantageous sporting situations. Therefore, physical sports practice can generate the experience of specific situations allowing the development of resilience [45]. In this case, the study developed by Ramírez-Granizo et al. [46] found that physical activity helps to increase academic performance, showing that students who obtained higher grades were those who showed higher levels of resilience.

With respect to the structural equation models, a negative relationship was observed between stress and resilience, and the same was true for resilience and burnout syndrome. Given these results, physical exercise promotes the development of people’s resilience [47]. In addition, it has been observed that students can increase the use of techniques for channeling disruptive states, such as yoga or tai chi, thus helping to improve concentration [48].

Likewise, a positive relationship was observed between burnout syndrome and stress levels. Very dissimilar results were found by Martínez-Rubio et al. [49], who stated that the practice of physical exercise can reduce stress levels and, therefore, channel burnout syndrome. Physical exercise brings numerous benefits to emotional development, helping to improve mood and attitude when facing tasks [6,50]. In addition, it was observed that a longer time spent preparing for a task leads to higher levels of burnout [51].

Lastly, a positive relationship was observed between the number of study hours and stress, and the same was true for study time and levels of resilience. In view of these results, Ramli et al. [52] affirmed that, when preparing for a particular test, the hours spent preparing can lead to negative emotions and foster states such as physical and emotional fatigue. However, resilience plays a fundamental role in the dedication to the preparation of a given test, as it was found that resilient people obtain better results when preparing for a given test, reflecting better results during the test [53].

## 5. Limitations and Future Perspectives

Despite the fact that this research fulfilled the established objectives, it also had a series of limitations, which are described below. The first limitation is that, as this was a cross-sectional study, it was only possible to establish the cause–effect relationships of the variables at that point in time. In addition, the questionnaires used, despite being validated and obtaining high degrees of reliability, presented an intrinsic error in the measurement.

In terms of future prospects, this research highlighted the importance of mental health in the complex process of access to the Spanish public teaching profession. In addition, the importance and benefits of increased weekly physical activity were also highlighted. For this reason, an intervention program is being developed in which, through the practice of physical activity, levels of resilience can be promoted in order to alleviate the disruptive states generated.

## 6. Conclusions

The descriptive analysis of the present research supports the hypothesis that physical activity helps to improve mental health, as lower levels of stress, emotional fatigue, and depersonalization were observed for those who were physically active. It was also shown that physical activity improves the resilience of participants, as higher scores were observed in all dimensions of this variable for those who exceeded 180 min of physical activity per week.

Regarding the structural equation models, it was observed that participants who complied with the recommendation of practicing more than 3 h of physical activity per week showed better relationships between hours of study and resilience, as well as between the academic variable and burnout. Similarly, it was also observed that participants who practiced less than 180 min of physical exercise per week showed better relationships between stress and burnout, as well as between resilience and burnout, than physically active participants.

## Figures and Tables

**Figure 1 behavsci-12-00329-f001:**
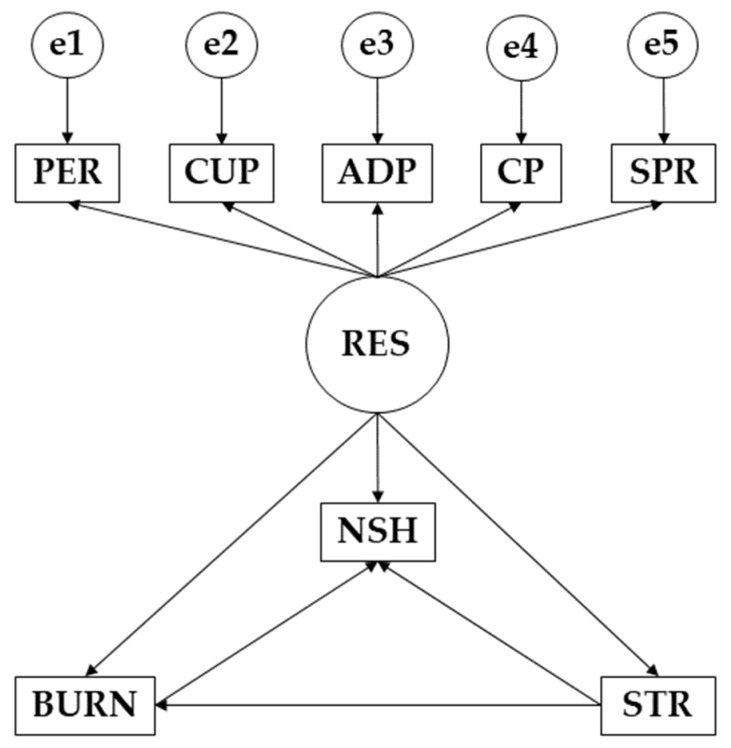
Structural equation model. **Note:** Resilience (RES); persistence/tenacity/self-efficacy (PER); control under pressure (CUP); adaptability and support networks (ADP); control and purpose (CP); spirituality (SP); number of study hours (NSH); burnout syndrome (BURN); stress (STR).

**Figure 2 behavsci-12-00329-f002:**
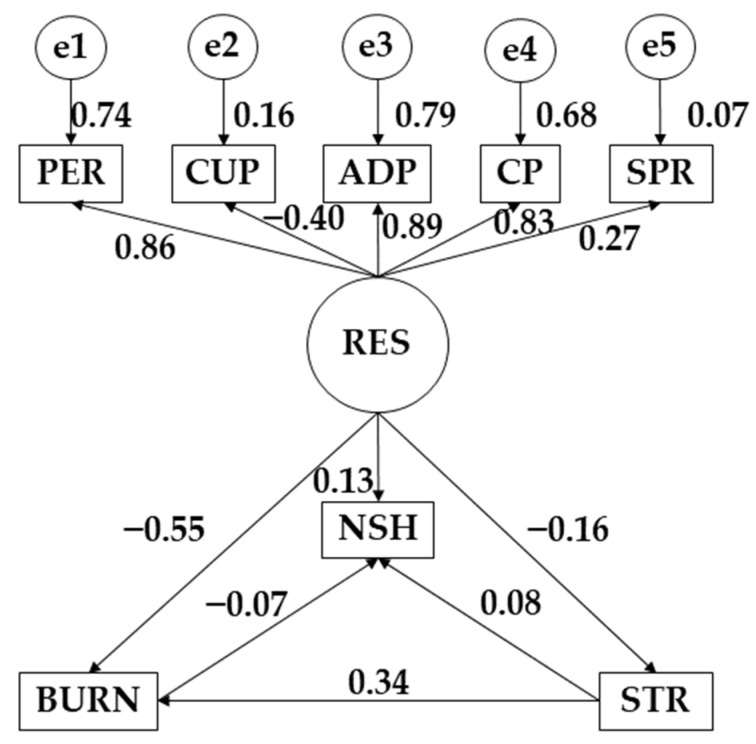
Structural equation model pertaining to participants who engaged in more than 3 h of physical activity a week. **Note:** Resilience (RES); persistence/tenacity/self-efficacy (PER); control under pressure (CUP); adaptability and support networks (ADP); control and purpose (CP); spirituality (SP); number of study hours (NSH); burnout syndrome (BURN); stress (STR).

**Figure 3 behavsci-12-00329-f003:**
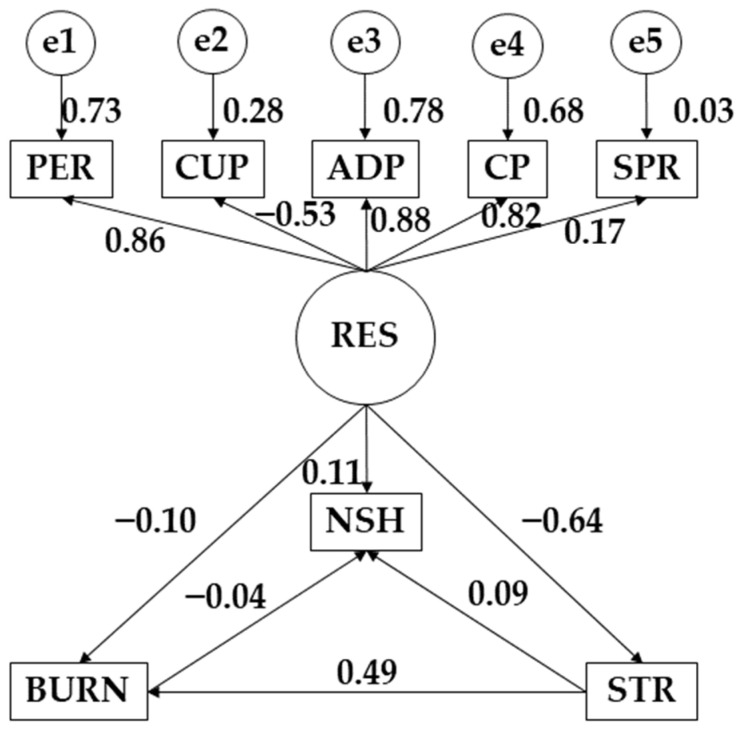
Structural equation model pertaining to participants who did not engage in more than 3 h of physical activity a week. **Note:** Resilience (RES); persistence/tenacity/self-efficacy (PER); control under pressure (CUP); adaptability and support networks (ADP); control and purpose (CP); spirituality (SP); number of study hours (NSH); burnout syndrome (BURN); stress (STR).

**Table 1 behavsci-12-00329-t001:** Comparative study of the sample.

		Levene Test					T-Test			ES (d)	95% CI
		*N*	M	SD	F	Sig	T	gl	*p*		
**STR**	**Yes**	2030	33.94	9.18	41.05	0.000	−12.936	41.12	≤0.05	0.402	[0.340; 0.464]
	**No**	2085	37.40	8.01							
**EE**	**Yes**	2030	36.43	8.64	41.73	0.000	−8.360	40.08	≤0.05	0.271	[0.210; 0.333]
	**No**	2085	38.61	7.44							
**DP**	**Yes**	2030	16.07	6.65	15.94	0.000	−4.359	41.82	≤0.05	0.135	[0.074; 0.196]
	**No**	2085	16.94	6.22							
**PR**	**Yes**	2030	25.20	8.11	1.86	0.172	8.372	41.11	≤0.05	0.261	[0.199; 0.322]
	**No**	2085	23.12	7.84							
**CP**	**Yes**	2030	2.65	0.75	2.935	0.087	10.165	40.10	≤0.05	0.327	[0.265; 0.388]
	**No**	2085	2.40	0.78							
**PER**	**Yes**	2030	2.39	0.89	8.991	0.003	−9.607	39.30	≤0.05	0.301	[0.239; 0.362]
	**No**	2085	2.65	0.84							
**CUP**	**Yes**	2030	2.79	0.64	1.216	0.270	10.753	40.77	≤0.05	0.354	[0.292; 0.415]
	**No**	2085	2.56	0.66							
**ADP**	**Yes**	2030	2.58	0.71	3.88	0.049	9.131	40.97	≤0.05	0.290	[0.228; 0.351]
	**No**	2085	2.37	0.74							
**SP**	**Yes**	2030	2.36	0.84	5.82	0.016	3.205	40.08	≤0.05	0.105	[0.043; 0.166]
	**No**	2085	2.27	0.88							

**Note:** Stress (STR); emotional exhaustion (EE); depersonalization (DP); personal realization (PR); control and purpose (CP); persistence/tenacity/self-efficacy (PER); control under pressure (CUP); adaptability and support networks (ADP); spirituality (SP).

**Table 2 behavsci-12-00329-t002:** Values obtained from the proposed structural equation model for participants doing more than 3 h of physical activity per week.

Associations between Variables	R.W.				S.R.W.
	Estimations	S.E.	C.R.	*p*	Estimations
STR←RES	−6.540	0.249	−26.247	***	−0.549
BURN←RES	−0.143	0.022	−6.399	***	−0.162
BURN ←STR	0.025	0.002	14.028	***	0.337
PER←RES	1.000				0.860
CUP←RES	−0.499	0.027	−18.201	***	−0.399
ADP ←RES	0.880	0.018	50.170	***	0.890
CP ←RES	0.918	0.020	45.741	***	0.827
SPR ←RES	0.359	0.030	12.094	***	0.272
NHS ←RES	0.451	0.098	4.614	***	0.131
NHS ←STR	0.024	0.008	3.056	**	0.085
NHS ←BURN	−0.291	0.095	−3.060	**	−0.074

**Note 1:** Regression weights (RW); standardized regression weights (SRW); standard error (SE); critical ratio (CR). **Note 2:** Resilience (RES); persistence/tenacity/self-efficacy (PER); control under pressure (CUP); adaptability and support networks (ADP); control and purpose (CP); spirituality (SP); number of study hours (NSH); burnout syndrome (BURN); stress (STR). **Note 3**: ** *p* ≤ 0.05; *** *p* ≤ 0.001.

**Table 3 behavsci-12-00329-t003:** Values obtained from the proposed structural equation model for participants doing less than 3 h of physical activity per week.

Associations between Variables	R.W.				S.R.W.
	Estimations	S.E.	C.R.	*p*	Estimations
STR←RES	−9.060	0.291	−31.133	***	−0.637
BURN ←STR	0.040	0.002	19.655	***	0.489
BURN←RES	−0.114	0.031	−3.746	***	−0.098
PER←RES	1.000				0.857
CUP←RES	−0.734	0.030	−24.872	***	−0.531
ADP ←RES	0.879	0.018	48.778	***	0.880
CP ←RES	0.909	0.020	44.532	***	0.822
SPR ←RES	0.216	0.030	7.141	***	0.165
NHS ←BURN	−0.139	0.084	−1.655	0.098	−0.044
NHS ←STR	0.024	0.008	2.899	**	0.094
NHS ←RES	0.417	0.116	3.592	***	0.113

**Note 1:** Regression weights (RW); standardized regression weights (SRW); standard error (SE); critical ratio (CR). **Note 2:** Resilience (RES); persistence/tenacity/self-efficacy (PER); control under pressure (CUP); adaptability and support networks (ADP); control and purpose (CP); spirituality (SP); number of study hours (NSH); burnout syndrome (BURN); stress (STR). **Note 3**: ** *p* ≤ 0.05; *** *p* ≤ 0.001.

## Data Availability

The data used to support the findings of current study are available from the corresponding author upon request.

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
