# Peer review of "Resilience, Stress, and Burnout Syndrome According to Study Hours in Spanish Public Education School Teacher Applicants: An Explanatory Model as a Function of Weekly Physical Activity Practice Time"

_behavsci, 2022, doi:10.3390/bs12090329_

Round 1

Reviewer 1 Report

There is a lack of literature review and that of theoretical background regarding phyisical activities. Thus, similar lack is seemed in the Discussion section, should be improved. 

There should be an operational definition of physical activity. Are the authors sure that the participants understood the same regarding the "physical activity"? 

The authors must explain why they chose three hours as a critical time. Are they sure that the participants did not fail in determining their time spend for physical activity? 

Author Response

REVIEWER 1

Comment 1

There is a lack of literature review and that of theoretical background regarding phyisical activities. Thus, similar lack is seemed in the Discussion section, should be improved. 

Response 1

Thank you very much for your comment. In this case both sections have been expanded in response to your suggestion.

Comment 2

There should be an operational definition of physical activity. Are the authors sure that the participants understood the same regarding the "physical activity"? 

Response 2

Thank you very much for your comment. In this case, following your suggestion, the definition of the practice of physical activity has been added to the body of the research.

Regarding the second part of your comment, the definition used in this research was specified in the question to make it easier for participants to understand the concept of "physical activity".

Comment 3

The authors must explain why they chose three hours as a critical time. Are they sure that the participants did not fail in determining their time spend for physical activity?

Response 3

Thank you very much for your comment and interest. In this case, as specified in the Procedure section, a literature review was previously carried out to determine the instruments with a higher degree of reliability. In this case, it was found that the research carried out by Arufe et al. (2019)1 used this form of data collection, considering that a subject who practices physical activity three or more hours a week outside is physically active, so this criterion has been used, coinciding with international recommendations.

In addition, to ensure that participants did not fail to complete this response, the definition of the concept of "physical activity" was provided.

1 Arufe-Giráldez, V.; Zurita-Ortega, F.; Padial-Ruz, R.; Castro-Sánchez, M. Association between Level of Empathy, Attitude towards Physical Education and Victimization in Adolescents: A Multi-Group Structural Equation Analysis. Int. J. Environ. Res. Public Health 201916, 2360. https://doi.org/10.3390/ijerph16132360

Reviewer 2 Report

Influence of Resilience, Stress and Burnout Syndrome on Study Hours in Spanish Public Elementary Education School Teacher Applicants. An Explanatory Model as a Function of Weekly Physical Activity Practice Time

This is an interesting contribution to the journal, but I would suggest some suggest some changes before it is fit for publication:

1.     Title: I would suggest authors to rewrite the title. “Influence” is appropriate for experimental or quasi-experimental research design, which is not the case. Also, it is too long and confusing.

2.     Abstract: Please adhere to the journal’s format. Also, provide more relevant information regarding the results.

3.     Introduction: More results from previous research with similar populations and measures is needed.

4.     An implications section would be helpful, focusing on mental health of future teachers and the importance of physical activity in curricula and educational policies.

Best wishes.

Author Response

REVIEWER 2

This is an interesting contribution to the journal, but I would suggest some suggest some changes before it is fit for publication:

Comment 1

Title: I would suggest authors to rewrite the title. “Influence” is appropriate for experimental or quasi-experimental research design, which is not the case. Also, it is too long and confusing.

Response 1

Thank you very much for your comment. In this case the title of the paper has been changed.

Comment 2

Abstract: Please adhere to the journal’s format. Also, provide more relevant information regarding the results.

Response 2

Thank you for your comment. In the abstract the results have been redrafted and adapted to the requested format.

Comment 3

Introduction: More results from previous research with similar populations and measures is needed.

Response 3

Thank you very much for your comment. In this case the introduction has been expanded in response to your suggestion.

Comment 4

An implications section would be helpful, focusing on mental health of future teachers and the importance of physical activity in curricula and educational policies.

Response 4

Thank you very much for your comment. The authors agree with your suggestion and therefore the section on limitations and future perspectives has been added as requested in your comment.

Reviewer 3 Report

Please review the following suggestions:

• Check the entire document and avoid redundancy, for example in the Abstract: the word process, helps; in the introduction: in order to, due to, etc.

• Check double spaces, for example, page 2 line 53, line 120

Author Response

REVIEWER 3

Please review the following suggestions:

Comment 1

Check the entire document and avoid redundancy, for example in the Abstract: the word process, helps; in the introduction: in order to, due to, etc.

Response 1

Thank you very much for your comment. The authors agree with your comment and redundancies have been removed.

Comment 2

Check double spaces, for example, page 2 line 53, line 120

Response 2

Thank you very much for your comment. The double spaces have been removed.

Round 2

Reviewer 1 Report

The new version is an improved one to be accepted.